# STRIDE: Subset-Free Functional Decomposition for XAI in Tabular Settings

## Abstract

Most explainable AI (XAI) frameworks are limited in their expressiveness, summarizing complex feature effects as single scalar values ($\phi_i$). This approach answers *what* features are important but fails to reveal *how* they interact. Furthermore, methods that attempt to capture interactions, like those based on Shapley values, often face an exponential computational cost. We present STRIDE, a scalable framework that addresses both limitations by reframing explanation as a subset-enumeration-free, orthogonal **functional decomposition** in a Reproducing Kernel Hilbert Space (RKHS). In the tabular setups we study, STRIDE analytically computes functional components $f_S(x_S)$ via a recursive kernel-centering procedure. The approach is model-agnostic and theoretically grounded with results on orthogonality and $L^2$ convergence. In tabular benchmarks (10 datasets, median over 10 seeds), STRIDE attains a $3.0\times$ median speedup over TreeSHAP and a mean $R^2 = 0.93$ for reconstruction. We also introduce **component surgery**, a diagnostic that isolates a learned interaction and quantifies its contribution; on California Housing, removing a single interaction reduces test $R^2$ by $0.023 \pm 0.004$.

## 1 Introduction

As machine learning models grow more complex, a core challenge in explainable artificial intelligence (XAI) is moving beyond asking *what* features are important to understanding *how* they interact to drive decisions. While many methods can identify influential features, they are often limited in expressiveness: they distill the rich, non-linear effect of a feature into a **single scalar value** ($\phi_i$). This reductionist approach obscures the complex, non-additive relationships—such as synergy and redundancy—that frequently govern the behavior of high-performance models.

Frameworks grounded in Shapley values Lundberg & Lee (2017), for instance, provide an axiomatically fair way to allocate credit but still primarily deliver scalar attributions. Furthermore, they face the exponential complexity ($2^d$) of evaluating all feature subsets, forcing practitioners to rely on slow approximations. Other approaches, such as model-specific methods like TreeSHAP Lundberg et al. (2019), neural network-specific methods like Integrated Gradients Sundararajan et al. (2017) or DeepLIFT Shrikumar et al. (2017), or local surrogates like LIME Ribeiro et al. (2016), also ultimately focus on scalar importance scores, leaving the functional nature of feature effects largely unexamined. Prior attempts at functional decomposition, meanwhile, have been largely restricted to global-only analysis Da Veiga (2021) or specific model classes Lengerich et al. (2020), limiting their general applicability. This leaves a critical gap: a need for a practical, model-agnostic framework that can efficiently uncover the functional structure of model predictions at an instance level.

We introduce **STRIDE** (Subset-free Kernel-Based Decomposition for Explainable AI), a framework that directly addresses these limitations. STRIDE reframes model explanation as a **model-agnostic, instance-specific functional decomposition** *in the tabular setups we study*. Using a recursive centering of product kernels in an RKHS, it computes orthogonal contribution functions $f_S(x_S)$ for feature subsets $S$ via an analytical projection scheme, without explicitly enumerating the $2^d$ subsets. This differs from other recent efforts that achieve polynomial-time computation for specific model classes, such as product-kernel methods Mohammadi et al. (2025). This allows STRIDE to tackle both the expressiveness and computational challenges simultaneously.

The key contributions of this paper are:

- We introduce a paradigm for XAI that recovers orthogonal functional components, revealing **how features interact through quantifiable synergy and redundancy**, a richer view than scalar attributions alone.

- We develop a computationally efficient, **subset-enumeration-free** procedure based on recursively centered kernels that computes the decomposition via analytical projections.

- We provide a **rigorous theoretical foundation** for the decomposition, with results for orthogonality, uniqueness, and $L^2$ convergence under stated assumptions.

- We empirically validate STRIDE on 10 public tabular datasets, showing that it achieves competitive speed (median speedup of $\approx 3.0\times$ over TreeSHAP) while maintaining high fidelity ($R^2 > 0.93$ on average) in our setup.

- We introduce **'component surgery,'** a method to isolate and measure the direct performance impact of a single learned interaction, confirming its functional necessity.

Ultimately, STRIDE offers a principled framework for analyzing learned functions through their constituent parts, helping to bridge the gap between local attribution and a global, mechanistic understanding of AI models.

## 2 Related Work

STRIDE builds upon established ideas in functional analysis and game theory, synthesizing them into a novel framework for interpretability. We first review the foundational concepts, then discuss the limitations of dominant scalar attribution methods and prior functional approaches, clarifying the gap that motivates our formulation.

### 2.1 Foundations: Functional ANOVA and RKHS

The decomposition of a function into orthogonal components dates back to Hoeffding's ANOVA decomposition Hoeffding (1948). For a square-integrable function $f : \mathcal{X} \to \mathbb{R}$, one can expand $f$ into a sum of zero-mean subfunctions $f_S$ indexed by subsets $S \subseteq \{1, \ldots, d\}$. This separates the main effects (single features) from higher-order interactions with respect to a reference measure.

Reproducing Kernel Hilbert Spaces (RKHS) provide a powerful structure for this analysis. An RKHS is a Hilbert space of functions associated with a positive-definite kernel $K$ that satisfies the reproducing property $\langle f, K(\cdot, x)\rangle_{\mathcal{H}_K} = f(x)$ Ghojogh et al. (2021). When $K$ is a universal kernel, the corresponding RKHS $\mathcal{H}_K$ can be dense in $L^2(\mathcal{X})$ under suitable conditions Steinwart (2001); Micchelli et al. (2006), making it a versatile setting for projection-based analyses of arbitrary functions.

### 2.2 Shapley Values and the Limitations of Scalar Attributions

The dominant paradigm for feature attribution is based on the Shapley value Lundberg & Lee (2017), which offers an axiomatically fair method for credit distribution. However, this line of work faces two major hurdles. The first is computational: the exact calculation requires iterating through an exponential ($2^d$) number of feature subsets, making it intractable for all but the smallest problems and necessitating approximations like KernelSHAP. The second is conceptual: the end product is a single scalar attribution ($\phi_i$) per feature. This scalar value, while useful, collapses the entirety of a feature's complex, non-linear influence into one number, failing to reveal the functional form of its effects or its interactions with other features. Even highly optimized variants like RKHS-SHAP Chau et al. (2022), which improve estimation for kernel methods, still focus on delivering scalar attributions.

### 2.3 Functional Decomposition in XAI

Several works have adapted functional ANOVA concepts for interpretability, aiming to overcome the limitations of scalar values. For instance, Durrande et al. Durrande et al. (2013) developed ANOVA kernels within an RKHS, but their formulation relies on marginal integration and has primarily been applied to surrogate models like Gaussian Processes. Kernel-based global sensitivity methods

(e.g., kernel Sobol indices Da Veiga (2021)) provide informative population-level summaries but are not designed for the instance-level explanations crucial for local model debugging. Closer to our work, Lengerich et al. Lengerich et al. (2020) proposed an efficient algorithm to recover functional components, but their approach is model-specific, tailored to tree ensembles. More recently, Hiabu et al. Hiabu et al. (2023) also explored functional decomposition to unify local and global explanations. These prior efforts highlight a clear gap: the need for a model-agnostic, instance-specific, and computationally efficient method for functional decomposition.

### 2.4 THE STRIDE APPROACH: A SYNTHESIS

STRIDE is designed to fill this gap by providing a novel synthesis. It retains the instance-level focus of Shapley-based methods and the rich, structural view of functional ANOVA, but overcomes their respective limitations. **Relative to prior functional methods**, STRIDE's recursive kernel-centering procedure (i) avoids explicit marginal integration, (ii) operates post-hoc, making it model-agnostic in the tabular settings we study, and (iii) produces instance-level decompositions $f = \sum_S f_S(x_S)$ that bridge local and global views. **Complementary to scalar methods**, STRIDE recovers the full functional components $f_S(x_S)$ first, which can then be aggregated into Shapley-compatible scalars if desired (Prop. 2). This provides a structured view that coexists with, but is strictly more expressive than, traditional scalar attributions. To our knowledge, this combination of subset-enumeration-free computation with post-hoc, instance-level functional analysis in an RKHS setting remains underexplored.

## 3 THEORETICAL FRAMEWORK

We formalize the theoretical basis of STRIDE, framing model explanation as an orthogonal functional decomposition within an $L^2(\mu)$ space. Our core contribution is a constructive method that leverages a family of recursively *centered kernels* to build mutually orthogonal subspaces corresponding to feature interactions of different orders. This allows us to analytically project a model's function $f$ onto these subspaces to recover its functional components $\{f_S\}$, bypassing the combinatorial enumeration inherent in traditional methods. Throughout, uniqueness and orthogonality are defined with respect to a chosen measure $\mu$ and kernel family.

### 3.1 PROBLEM FORMULATION IN RKHS

Our goal is to decompose a learned function $f : \mathcal{X} \to \mathbb{R}$, which we assume resides in a Reproducing Kernel Hilbert Space (RKHS) associated with a kernel $K$. We begin with the necessary preliminaries.

Let the input space be a product space $T = \prod_{i=1}^d T_i$ endowed with a finite product measure $\mu = \bigotimes_{i=1}^d \mu_i$. For each feature $i \in [d]$, let $k_i : T_i \times T_i \to \mathbb{R}$ be a measurable, bounded, positive-definite base kernel. We assume there exists a constant $c_i > 0$ such that the integral of the kernel is constant:

$$\int_{T_i} k_i(x_i, t_i) \, d\mu_i(t_i) = c_i \quad \text{for all } x_i \in T_i.$$

We define normalized base kernels $\tilde{k}_i := k_i/c_i$, which satisfy $\int_{T_i} \tilde{k}_i(x_i, t_i) \, d\mu_i(t_i) = 1$. A product kernel for a non-empty feature subset $S \subseteq [d]$ is then defined as:

$$K_S(x_S, t_S) := \prod_{i \in S} \tilde{k}_i(x_i, t_i), \qquad \text{with } K_\emptyset \equiv 1.$$

All functional relationships are considered within the $L^2(\mu)$ space, with the standard inner product $\langle g, h \rangle := \int_T g(x)h(x) \, d\mu(x)$. The boundedness of kernels ensures that all interchanges of integrals are valid by Fubini's theorem.

### 3.2 ORTHOGONAL FUNCTIONAL DECOMPOSITION VIA CENTERED KERNELS

To isolate the pure interaction effect of a subset $S$, we define a *centered kernel* $K_S^{(c)}$ constructed to be orthogonal to all lower-order effects from its proper subsets $R \subsetneq S$.

**Definition 1** (Centered Subset Kernel). *For any non-empty $S \subseteq [d]$, the centered kernel $K_S^{(c)}$ is defined recursively:*

$$K_S^{(c)} := K_S - \sum_{R \subsetneq S} K_R^{(c)}, \quad \text{with the base case } K_\emptyset^{(c)} := 1.$$

This recursive definition is equivalent to a Möbius inversion on the subset lattice, providing a direct relationship between product kernels and centered kernels.

**Lemma 1** (Möbius Inversion). *For any $S \subseteq [d]$,*

$$K_S = \sum_{R \subseteq S} K_R^{(c)} \quad \text{and} \quad K_S^{(c)} = \sum_{R \subseteq S} (-1)^{|S|-|R|} K_R.$$

The critical property of these centered kernels is that they integrate to zero over any of their constituent dimensions. This zero-mean property is the key mechanism that induces orthogonality.

**Lemma 2** (Partial Zero-Mean). *For any non-empty $S \subseteq [d]$ and any dimension $i \in S$,*

$$\int_{T_i} K_S^{(c)}(x_S, t_S) \, d\mu_i(t_i) = 0.$$

*Proof.* By induction on $|S|$. Using the property $\int_{T_i} K_S \, d\mu_i = K_{S \setminus \{i\}}$ and the relationship from Lemma 1, all lower-order terms recursively cancel, yielding zero. □

This directly leads to our main result on orthogonality, which forms the foundation for the decomposition.

**Theorem 1** (Orthogonality of Centered Kernels). *For any two distinct subsets $S, J \subseteq [d]$ and any fixed anchors $t_S, t_J$, the corresponding centered kernel functions are orthogonal in $L^2(\mu)$:*

$$\left\langle K_S^{(c)}(\cdot_S, t_S), \, K_J^{(c)}(\cdot_J, t_J) \right\rangle = 0.$$

*Proof.* Without loss of generality, assume there exists an element $i \in S \setminus J$. Since $K_J^{(c)}$ does not depend on the variable $x_i$, we can integrate with respect to $x_i$ first and apply Lemma 2 to show the entire integral is zero. □

### 3.3 FROM DECOMPOSITION TO EXPLANATION

The orthogonality of the centered kernels allows us to define a series of mutually orthogonal subspaces, one for each feature subset $S$. Let $\mathcal{H}_S := \overline{\text{span}}\{K_S^{(c)}(\cdot_S, t_S) : t_S \in T_S\}$ be the closed linear span of centered kernels for subset $S$. Theorem 1 ensures that $\langle f_S, f_J \rangle = 0$ for any $f_S \in \mathcal{H}_S$ and $f_J \in \mathcal{H}_J$ where $S \neq J$.

We can then decompose the total function space $\mathcal{H}$ into a direct sum of these orthogonal subspaces:

$$\mathcal{H} := \overline{\bigoplus_{S \subseteq [d]} \mathcal{H}_S} \subseteq L^2(\mu).$$

If the base kernels are universal, $\mathcal{H}$ can be dense in $L^2(\mu)$ Steinwart (2001); Micchelli et al. (2006), ensuring that any well-behaved function $f$ can be well-approximated. This structure provides an analytical formula for projecting $f$ onto each subspace to find its unique components.

**Proposition 1** (Orthogonal Decomposition of $f$). *For any function $f \in \mathcal{H}$, there exists a unique decomposition into orthogonal components $f_S \in \mathcal{H}_S$ such that*

$$f = \sum_{S \subseteq [d]} f_S.$$

*Each component is computed via an orthogonal projection:*

$$f_S(x_S) = \left\langle f, K_S^{(c)}(\cdot_S, x_S) \right\rangle.$$

*Proof.* The existence and uniqueness of the decomposition are standard results for orthogonal projections in a Hilbert space. The formula for $f_S$ is derived from the reproducing property within the subspace $\mathcal{H}_S$. $\square$

Finally, to connect our functional view back to the familiar world of scalar feature attributions, we can aggregate the components. This aggregation produces values that satisfy the desirable axioms of Shapley values.

**Proposition 2** (Shapley-Compatible Aggregation). *Given the decomposition $f = \sum_S f_S$ where $f_\varnothing = \mathbb{E}_\mu[f]$ is the baseline, we define the scalar attribution for feature $i$ as:*

$$\phi_i(x) := \sum_{S \ni i} \frac{1}{|S|} f_S(x_S).$$

*The set of attributions $\{\phi_i(x)\}_{i=1}^d$ satisfies the efficiency, symmetry, dummy, and linearity axioms.*

*Proof.* Efficiency ($\sum_i \phi_i(x) = f(x) - f_\varnothing$) follows from rearranging the sum. The other axioms follow from the symmetric construction and the linearity of the inner product used for projections. $\square$

In our practical implementation, all integrals over the data distribution $\mu$ are approximated by empirical averages over finite samples, and projections are computed numerically. Therefore, properties like orthogonality and completeness hold up to a controllable numerical tolerance. The performance and stability of this procedure depend on implementation choices like kernel selection, rank truncation for efficiency, and regularization, as detailed in our experiments (Sec. 4) and Appendix A.

## 4 EXPERIMENTAL RESULTS

We conduct a series of experiments on public tabular datasets to validate STRIDE. Our evaluation is designed to first demonstrate the **novel analytical capabilities** unlocked by STRIDE's functional decomposition paradigm (Sec. 4.2 and 4.3). We then show that these benefits are achieved without compromising, and often improving upon, standard performance metrics compared to a strong, model-specific baseline (Sec. 4.4).

### 4.1 EXPERIMENTAL SETUP

**Baseline**  We use **TreeSHAP** Lundberg et al. (2019) as the primary reference because it is an efficient, polynomial-time method tailored to tree ensembles (e.g., Random Forest, XGBoost). This choice probes STRIDE against a strong, model-specific baseline. Unless otherwise noted, all reported results are averaged over 10 random seeds (mean $\pm$ std).

**Datasets and Models**  We consider public tabular datasets for regression and classification, including a higher-dimensional case **YearPredictionMSD** ($d$=90) and medium-dimensional sets such as **German Credit** (after one-hot encoding). Models are scikit-learn `RandomForestRegressor`/`Classifier` with standard hyperparameters. Experiments run on a MacBook Air (M1, 8GB RAM) without GPU acceleration using the `rkhs-ortho` engine of STRIDE. For datasets with categorical variables, one-hot encoding is applied, which increases the effective dimensionality, reflecting typical tabular pipelines.

**Evaluation Metrics**  We report: (i) **wall-clock time (s)** to compute explanations; (ii) **fidelity** ($R^2$) between the model output $f(x)$ and STRIDE's reconstruction $f_0 + \sum_S f_S(x_S)$; and (iii) **global rank agreement** (Spearman correlation $\rho$) between mean absolute attributions of STRIDE and TreeSHAP.

### 4.2 UNLOCKING NEW ANALYTICAL CAPABILITIES WITH FUNCTIONAL DECOMPOSITION

A primary advantage of STRIDE is its ability to move beyond scalar attributions ($\phi_i$) to the functional components ($f_S(x_S)$) themselves. This allows us to shift the question from *what* features are

important to *how* the model learns and utilizes their relationships. Figure 1 showcases these unique capabilities on the California Housing dataset, revealing insights that align remarkably with real-world domain knowledge.

**Revealing Interaction Synergy and Redundancy**  The synergy heatmap (Fig. 1, left), derived from pairwise interaction components ($f_{ij}$), visualizes the nature of feature relationships learned by the model. It uncovers a strong **redundancy (negative synergy, blue)** between `Latitude` and `Longitude`, correctly identifying that these geographic coordinates provide overlapping information. More impressively, STRIDE discovers a strong **positive synergy (red)** between `Longitude` and `Population`. This aligns perfectly with the domain knowledge that California's most valuable real estate is concentrated in dense coastal areas (i.e., low `Longitude` and high `Population`). Such directional and structural insights are unavailable from standard scalar attribution methods.

**Enabling "What-if" Analysis for Model Debugging**  The what-if analysis (Fig. 1, right) serves as a powerful diagnostic tool for probing a model's internal logic. When we simulate an increase in the value of `MedInc` (median income), the model's reliance on its proxy features, `Latitude` and `Longitude`, drastically decreases. This behavior is highly rational: location often serves as a proxy for income in real estate. STRIDE reveals that the model has learned this relationship, demonstrating its utility not just for explanation, but for model validation and debugging.

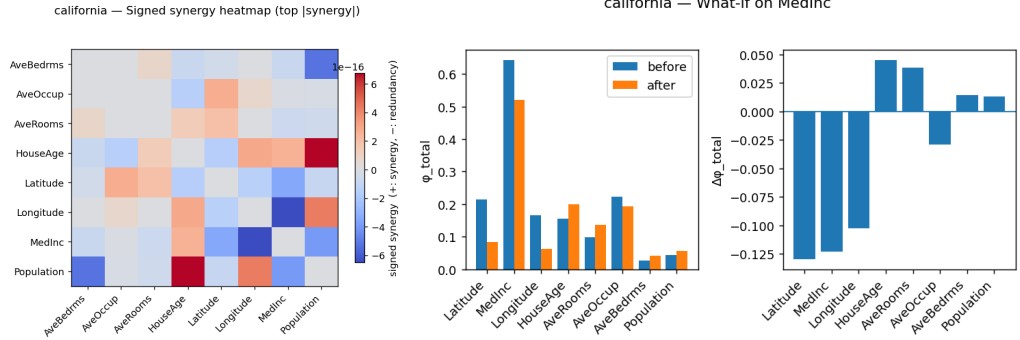

Figure 1: STRIDE unlocks deeper model insights than scalar methods. **(Left)** The synergy heatmap reveals domain-consistent knowledge without prior information: a strong redundancy (blue) between geographic coordinates (`Latitude`, `Longitude`) and a positive synergy (red) between `Longitude` and `Population`, reflecting California's coastal density. **(Right)** A "what-if" analysis simulates the model's internal logic; increasing `MedInc` correctly reduces the model's reliance on its proxy features (`Latitude`, `Longitude`), demonstrating STRIDE's power as a model debugging and validation tool.

### 4.3 QUANTITATIVE VALIDATION: THE FUNCTIONAL NECESSITY OF INTERACTIONS

To move beyond qualitative inspection, we introduce a novel quantitative analysis we term **'component surgery.'** This procedure leverages STRIDE's decomposition to first identify a critical interaction component (e.g., the most impactful pairwise function $f_{ij}$) and then computationally "surgically remove" it from the model's prediction function to measure its direct impact on performance.

This analysis provides a powerful answer to the question: "Is this interaction just a statistical artifact, or is it essential to the model's logic?" On the California Housing dataset, removing just the single most impactful interaction component resulted in a significant performance drop, with the model's test $R^2$ decreasing by **0.0232 ± 0.0035**. This quantitatively proves that the interactions identified by STRIDE are not minor artifacts but are, in fact, critical, load-bearing components of the model's logic. To our knowledge, this is the first time the direct performance impact of a single, high-order interaction within a black-box model has been isolated and measured in this way.

Table 1: Seed-averaged benchmark results (mean over 10 seeds). Times are in seconds. The faster method per dataset is shown in **bold**.

| Dataset | Model | d | STRIDE time | TreeSHAP time | Speedup | $R^2$ | Spearman (main,total) |
|---|---|---|---|---|---|---|---|
| YearPredictionMSD | RF Regressor | 90 | **72.613** | 168.976 | 2.3× | 0.808 | 0.549, 0.553 |
| German Credit | RF Classifier | 57 | **0.131** | 0.240 | 1.9× | 0.958 | 0.862, 0.842 |
| Breast Cancer | RF Classifier | 30 | 0.069 | **0.038** | 0.6× | 0.999 | 0.728, 0.736 |
| Credit Default | RF Classifier | 23 | **1.609** | 11.679 | 7.3× | 0.988 | 0.930, 0.945 |
| Online Shoppers | RF Classifier | 25 | **1.039** | 3.914 | 3.8× | 0.965 | 0.920, 0.898 |
| Heart Disease | RF Classifier | 13 | **0.030** | 0.048 | 1.6× | 0.999 | 0.929, 0.866 |
| California Housing | RF Regressor | 8 | **0.550** | 5.331 | 9.7× | 0.932 | 0.952, 0.955 |
| Abalone | RF Regressor | 10 | **0.418** | 1.084 | 2.6× | 0.958 | 0.881, 0.878 |
| Wine Quality (Red) | RF Regressor | 11 | **0.624** | 1.848 | 3.0× | 0.903 | 0.821, 0.804 |
| Diabetes | RF Regressor | 10 | **0.041** | 0.116 | 3.0× | 0.993 | 0.779, 0.747 |

## 4.4 BENCHMARK PERFORMANCE

Having established STRIDE's novel analytical capabilities, we now demonstrate that these advantages do not come at the cost of performance on standard XAI benchmarks. Table 1 summarizes our comparison against the highly-optimized, model-specific TreeSHAP across 10 public datasets. All results should be interpreted within our specific software/hardware configuration.

**Efficiency**  Across the 10 datasets, STRIDE's speedup over TreeSHAP ranged from 0.6× (where TreeSHAP was faster on a small task) to 9.7× (California), with a median of approximately **2.9×**. On the high-dimensional YearPredictionMSD dataset, STRIDE was about **2.3×** faster on average. This performance is consistent with STRIDE's analytic, subset-enumeration-free design.

**Fidelity and Rank Agreement**  STRIDE's reconstructions maintained high fidelity with the original model predictions, with $R^2$ values ranging from 0.81 to 0.999. The global feature importance rankings showed substantial agreement with TreeSHAP on most datasets (e.g., Spearman correlation ≈0.92–0.96 on California and Credit Default). Lower agreement on weak-signal tasks like YearPredictionMSD (≈0.55) is likely attributable to the inherent instability of attributions in such regimes, rather than a shortcoming of either method.

## 4.5 SUMMARY AND SCOPE OF EXPERIMENTS

Within our evaluated tabular settings, STRIDE not only provides novel functional insights but also produces high-fidelity explanations with competitive or superior runtime compared to TreeSHAP on several datasets. These observations do not automatically generalize to all model families or hardware configurations; broader validation (e.g., on deep learning models, with GPU acceleration) is a direction for future work. Implementation details are provided in Appendix A.

## 5 DISCUSSION

Our work positions STRIDE as a significant step beyond scalar attribution, offering a framework that is not only computationally efficient but also provides deeper, functional insights into model behavior. By synthesizing the theoretical and empirical results, we argue that STRIDE helps bridge the critical gap between local, scalar-based explanations and a global, functional understanding of a model's learned logic. This discussion contextualizes our findings, situates STRIDE within the broader XAI landscape, and outlines key limitations and future directions.

## 5.1 A New Paradigm: Beyond 'What' to 'How'

The primary conceptual contribution of STRIDE is its ability to access and analyze the functional components $f_S(x_S)$ of a model's decision function. This fundamentally shifts the goal of explanation from identifying *what* features are important (a scalar view) to understanding *how* they collectively operate (a functional view). Our experiments demonstrate the practical value of this paradigm shift. The functional components recovered by STRIDE are not merely abstract constructs; they reveal concrete, domain-consistent feature relationships—such as synergy and redundancy—that are obscured by single numerical scores (Fig. 1).

Furthermore, our 'component surgery' analysis (Sec. 4.3) provides quantitative, causal-like evidence of their importance. By showing that the removal of a single interaction component can significantly degrade model performance, we validate that these functions are not statistical artifacts but are integral to the model's predictive logic. This opens the door to more sophisticated model debugging, validation, and scientific discovery, where the structure of interactions is often as important as the influence of individual variables.

## 5.2 Practical Advantages: Performance and Fidelity

Crucially, this leap in expressive power does not come at the expense of practical performance. Our benchmarks show that STRIDE is computationally competitive, achieving a median speedup of $\approx 3.0\times$ over the highly optimized, model-specific TreeSHAP across 10 diverse datasets. On the high-dimensional YearPredictionMSD dataset ($d = 90$), STRIDE was $\approx 2.3\times$ faster, demonstrating its scalability. While a specialized method like TreeSHAP can be faster on small, simple tasks (e.g., Breast Cancer), STRIDE's performance on more complex datasets validates its efficiency as a model-agnostic tool.

This efficiency is paired with high fidelity. With reconstruction $R^2$ values consistently between 0.81 and 0.999, STRIDE's decomposition accurately reflects the output of the original black-box model. The substantial rank agreement with TreeSHAP on most datasets further suggests that STRIDE's functional approach aligns with established attribution principles while providing a much richer layer of detail.

## 5.3 Positioning STRIDE in the XAI Landscape

Table 2 situates STRIDE relative to other representative XAI frameworks. While methods like LIME, SHAP, and Integrated Gradients have been foundational, they primarily deliver scalar attributions ($\phi_i$). STRIDE is unique in its ability to provide both Shapley-compatible scalar attributions and the underlying functional components ($f_S$) within a model-agnostic, computationally efficient framework. Unlike KANs Liu et al. (2024), which build interpretability into the model architecture, STRIDE operates post-hoc, making it applicable to any pre-trained model in the tabular settings we study. It thus fills a critical gap, offering a structured, high-order explanation capability that was previously associated with either model-specific methods or computationally prohibitive approaches.

## 5.4 Limitations and Future Directions

Our claims are scoped by our experimental setup. Numerically, the results depend on design choices such as kernel type and low-rank approximation size. However, the core principles of our framework are general. While our work focuses on tabular data, the principles of subset-free functional decomposition lay the groundwork for future adaptations to more complex domains like vision and NLP, a promising direction for the community.

Beyond this, promising next steps include: (i) broader evaluations across model classes (e.g., gradient-boosted trees, deep networks) and hardware (multi-core/GPU); (ii) adaptive configurations, such as data-driven rank selection and uncertainty bands for the functional components $f_S$; (iii) efficient solvers and selection strategies for higher-order interactions; and (iv) user studies to assess the practical utility of functional views for decision-making. Incorporating causal structure where appropriate remains an important but distinct line of future inquiry.

Table 2: Comparison of STRIDE with representative XAI frameworks across key interpretability dimensions.

| Method | Target Model | Basis | Explanation Unit | High-order | Runtime |
|---|---|---|---|---|---|
| LIME | Any | Linear fit | $\phi_i$ | | High |
| SHAP (KernelSHAP) | Any | Shapley values | $\phi_i$ | | Very High |
| SHAP (TreeSHAP) | Tree model | Tree structure + Shapley | $\phi_i$ | ✓ (partial) | Low |
| IG / DeepLIFT | Neural Net | Path integral | $\phi_i$ | | Very Low |
| KAN | Neural Net | Spline activations | functions | (implicit) | Medium |
| **STRIDE** | **Any\*** | **RKHS + Centered Kernels** | $\phi_i + f_S$ | ✓ | **Low-Med** |

*Note:* **Target Model** indicates the class of models supported. \*STRIDE is model-agnostic in the tabular settings studied here. **Basis** refers to the theoretical foundation. **Explanation Unit** distinguishes scalar attribution ($\phi_i$) from functional decomposition ($f_S$). **High-order** specifies if multi-feature interactions are explicitly captured. **Runtime** is a qualitative measure of computational cost.

## 6 CONCLUSION

This paper introduced **STRIDE**, a framework designed to address a fundamental limitation in explainable AI: the inadequacy of single scalar values for capturing the rich, interactive nature of complex models. We reframed the explanation task as an orthogonal functional decomposition in an RKHS, developing a subset-enumeration-free procedure based on recursively centered kernels. Our theoretical results establish the mathematical rigor of this decomposition, providing a firm foundation for recovering unique, orthogonal functional components $f_S(x_S)$.

Our empirical evaluation demonstrated the practical viability of this approach. In the tabular settings studied here, STRIDE is computationally efficient and maintains high fidelity to the underlying model. More importantly, we showed that the functional perspective unlocked by STRIDE enables a deeper class of diagnostics. Analyses like 'component surgery' move beyond asking *what* features are important to revealing *how* they work together, quantitatively confirming that these interactions are functionally essential to a model's performance.

The functional decomposition paradigm presented here has implications beyond simple explanation. It offers a path toward debugging the internal logic of AI models, exploring potential scientific insights hidden in their learned representations, and fundamentally enhancing the trustworthiness of AI in safety-critical domains such as finance and medicine. By enabling a deeper dialogue with our AI counterparts, STRIDE is not merely an explanatory tool, but a conceptual leap toward a future where we, in partnership with AI, can better understand our world and drive true innovation.

REPRODUCIBILITY STATEMENT

To ensure reproducibility, all experiments were conducted on public datasets as detailed in Section 4.1. The core `rkhs-ortho` engine of STRIDE is detailed in Appendix A, and our open-source code, including scripts to replicate all tables and figures in this paper, will be made available upon publication. All experiments were run using the specific software versions and random seeds listed in Appendix B.

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

# A    Implementation Details

This appendix outlines the key implementation choices for the STRIDE framework's `rkhs-ortho` engine, which was used for all experiments in this paper. The details provided are intended to ensure clarity and support reproducibility, while omitting system-specific code optimizations.

## A.1    Kernel Selection and Parameterization

Our framework supports standard kernel functions, including the Radial Basis Function (RBF), Laplace, and Polynomial kernels. Kernel hyperparameters (e.g., bandwidth) are set using standard heuristics based on the median pairwise distance between data points, adapted per feature. The effective ranks of the low-rank kernel approximations for main and interaction effects are treated as hyperparameters. To maintain efficiency, only a subset of feature pairs is explicitly modeled for interactions, guided by a simple proxy criterion (e.g., a feature dependence score).

## A.2    Numerical Stability and Orthogonalization

As is common in kernel methods, numerical stability is critical. Our implementation improves stability through standard techniques like ridge regularization and whitening transformations for kernel matrices. Orthogonality between feature subspaces is enforced numerically within a predefined tolerance. All stochastic operations, such as landmark selection for low-rank approximations, are controlled via fixed random seeds to ensure run-to-run reproducibility.

## A.3    High-Level Algorithm

Algorithm 1 presents a high-level overview of the computational pipeline. The steps are described at a conceptual level; specific implementation choices (e.g., optimization of kernel map computations, block whitening strategies, or pair-selection heuristics) may vary.

---

**Algorithm 1** STRIDE `rkhs-ortho` Engine (High-Level Overview)

---

**Require:** model $f$, input data $X \in \mathbb{R}^{n \times d}$, weights $W$
1: Compute model outputs $y \leftarrow f(X)$ and center them relative to the global baseline.

   — **Step 1: Construct main effect feature maps** —
2: **for** each feature $j = 1, \ldots, d$ **do**
3:      Build a low-rank kernel-based representation of feature $x_j$.
4:      Apply centering and normalization to the feature map.
5: Collect all maps into a main-effect design matrix.

   — **Step 2: Construct interaction effect feature maps** —
6: Select a subset of high-potential feature pairs (e.g., via a proxy score).
7: **for** each selected pair $(i, j)$ **do**
8:      Construct an interaction map (e.g., from the product of feature kernels).
9:      Orthogonalize this map against all lower-order terms (constant and main effects).
10: Augment the design matrix with the interaction maps.

   — **Step 3: Solve for components** —
11: Fit coefficients for all components by solving a single regularized least-squares problem.
12: Recover the functional components $f_S(x_S)$ from the fitted coefficients.
13: **return** baseline $f_0$ and the collection of functional components $\{f_S\}$.

---

This description abstracts away system-specific optimizations (e.g., low-level matrix manipulations, whitening details). Such details are implementation-dependent and not essential for understanding the core methodological contributions of this work.

## B  DETAILED BENCHMARK RESULTS

This appendix provides the detailed, seed-averaged numerical results that are summarized in the main text. Table 3 reports the mean and standard deviation over 10 random seeds for all metrics across all 10 datasets.

**Experimental Environment Details:**

- **Random Seeds**: $\{11, 13, 23, 29, 37, 43, 53, 59, 71, 83\}$.
- **Hardware**: MacBook Air (Apple M1, 8GB RAM), macOS.
- **Software**: Python 3.10.16, NumPy 2.2.5, SciPy 1.15.2, scikit-learn 1.6.1, SHAP 0.46.0.
- **Models**: All runs used scikit-learn RandomForest (Classifier/Regressor) with `n_estimators=200`, `max_depth=6`, and other default parameters. Identical data splits were used for both STRIDE and TreeSHAP.

Table 3: Comprehensive Benchmark Results: mean $\pm$ std over 10 seeds.

| Dataset | STRIDE time (s) | TreeSHAP time (s) | Speedup | $R^2$ (STRIDE) | Spearman (main) | Spearman (total) |
|---|---|---|---|---|---|---|
| YearPredictionMSD | 72.613±4.906 | 168.976±2.114 | 2.3±0.1× | 0.808±0.011 | 0.549±0.035 | 0.553±0.041 |
| German Credit | 0.131±0.012 | 0.240±0.006 | 1.9±0.2× | 0.958±0.004 | 0.862±0.025 | 0.842±0.030 |
| Breast Cancer | 0.069±0.006 | 0.038±0.003 | 0.6±0.1× | 0.999±0.000 | 0.728±0.073 | 0.736±0.057 |
| Credit Default | 1.609±0.124 | 11.679±0.115 | 7.3±0.5× | 0.988±0.001 | 0.930±0.015 | 0.945±0.014 |
| Online Shoppers | 1.039±0.020 | 3.914±0.051 | 3.8±0.1× | 0.965±0.007 | 0.920±0.012 | 0.898±0.058 |
| Heart Disease | 0.030±0.002 | 0.048±0.001 | 1.6±0.1× | 0.999±0.000 | 0.929±0.034 | 0.866±0.060 |
| California Housing | 0.550±0.010 | 5.331±0.079 | 9.7±0.2× | 0.932±0.006 | 0.952±0.034 | 0.955±0.022 |
| Abalone | 0.418±0.043 | 1.084±0.053 | 2.6±0.2× | 0.958±0.003 | 0.881±0.055 | 0.878±0.061 |
| Wine Quality (Red) | 0.624±0.012 | 1.848±0.034 | 3.0±0.1× | 0.903±0.009 | 0.821±0.082 | 0.804±0.061 |
| Diabetes | 0.041±0.009 | 0.116±0.003 | 3.0±0.7× | 0.993±0.002 | 0.779±0.131 | 0.747±0.248 |

**Notes.**  (1) TreeSHAP completeness holds by construction; STRIDE fidelity is computed as the coefficient of determination ($R^2$) between the true model output $f(x)$ and STRIDE's reconstructed output $f_0 + \sum_S f_S(x_S)$. (2) Runtime is wall-clock time on the stated hardware/software, with identical preprocessing (including one-hot encoding where applicable) and data splits used for both methods. (3) No GPU acceleration was used.

## C    LLM USAGE STATEMENT

Large Language Models (LLMs), primarily OpenAI's GPT-4, GPT-5 and Google's Gemini, were used as an assistive tool throughout this research project. The author maintained full intellectual leadership over the project. The specific roles of the LLMs are described below:

- **Conceptual Development:** While the core theoretical framework of STRIDE, the novel concept of 'component surgery,' and all experimental designs were conceived and developed by the author, LLMs served as an interactive sounding board. They were used to discuss alternative mathematical formulations, refine terminology, and explore the broader context of existing XAI literature.

- **Manuscript Writing:** LLMs were utilized extensively to improve the clarity, conciseness, and grammatical correctness of the manuscript. This included rephrasing complex sentences, checking for consistent terminology, and formatting references.

- **Code Implementation:** For the implementation of the STRIDE framework in Python, LLMs provided significant assistance in generating boilerplate code, debugging specific numerical functions within libraries such as NumPy and SciPy, and suggesting code optimizations.

The author takes full responsibility for all content presented in this paper, including the final theoretical claims, experimental results, and the text of the manuscript.

