# OpenReview forum: "STRIDE: Subset-Free Functional Decomposition for XAI in Tabular Settings"
_ICLR.cc/2026/Conference — ICLR 2026 Conference Withdrawn Submission_

### Official Review · Reviewer_SVKr · 2025-10-22

**Soundness:** 3
**Presentation:** 3
**Contribution:** 2
**Rating:** 2
**Confidence:** 4

**Summary:**

This work frames post-hoc explainability of Machine Learning model predictions as a functional decomposition problem. It proposes a new technique to compute the decomposition of a model's output: 1) define a surrogate model based on orthogonalized kernels to model main effects $f_i$ and interactions $f_{ij}$ 2) fit this surrogate to estimate the original model's output 3) Optionally aggregate main effects and interactions to get SHAP values. The approach evaluated qualitatively on California Housing and compared quantitatively with TreeSHAP on ten tabular datasets.

**Strengths:**

The paper is well written and easy to follow.

To the best of my knowledge, **Proposition 1** is a novel theoretical result that highlights an interesting means of computing functional decompositions.

The **Component Surgery** method discussed in Section 4.3 is a very original and sound way of determining whether the interactions learned by a predictive model are spurious or truly present in the data.

Quantitative comparisons with TreeSHAP were done on a representative set of tabular data (regresssion/classification, small/medium/large number of features).

**Weaknesses:**

## Motivation

Section 2.2 motivates the need for an alternative method to SHAP (KernelSHAP, TreeSHAP etc.) because they only return a single attribution $\phi_i$ per input feature $i=1, 2, \ldots,d $. This may hide the presence of feature interactions (synergies, redundancies) in the model and or the data. Functional decomposition is proposed as an alternative that assigns importance to individual features **and** their interactions. While I agree that functional decompositions are more informative/useful and SHAP value (based on my personal experience), the paper does not demonstrate it to the reader. This is because STRIDE is both advertised as a functional decomposition algorithm and an alternative mean of computing Shapley Values (via Proposition 2). Since the experiments comparing SHAP to STRIDE are far more extensive (10 datasets) than the experiments studying the functional decomposition (qualitative examples on 1 dataset), STRIDE mostly comes out as a new SHAP estimate.

To remedy this issue, it would help to extend section 4.2 with clear qualitative examples of misleading explanations by SHAP and show that the functional decomposition of STRIDE does not mislead. These examples could even be synthetic if necessary.

## Comparison with other Functional Decomposition Algorithms

STRIDE is extensively compared with TreeSHAP, but not with existing methods that perform functional decomposition. Therefore, the benefits of using a basis of orthogonal kernels to compute the functional decomposition are not clear. Comparing the runtime and $R^2$ reconstruction error with existing functional decomposition methods (such as [1]) would highlight the merits of STRIDE.

## Lack of Technical Details

The manuscript is lacking details regarding how the functional decomposition is computed.

First, Proposition 1 suggests that the STRIDE computes the components $f_S(x_S)$ by computing a scalar product between $f$ and the centered kernel. However, Appendix A clarifies that STRIDE actually defines bases for the spaces $\mathcal{H}\_S$ and solves a regression problem
$$\min\_{f\_S\in \mathcal{H}\_S: S\subseteq [d] : |S|\leq 2}\mathbb{E}[(f(X) - \sum\_{S\subseteq [d] : |S| \leq 2} f\_S(X_S)^2)]$$
to find the decomposition. Is Proposition 1 used in the actual implementation? If not, the formulation as a least-square should instead be presented in Section 3.

Second, the implementation restricts the interactions to $|S|\leq 2$ (Algorithm 1 in Appendix A). Why is this restriction imposed apriori and not based on the data? The competing method [1] can identify a three-way interactions between
hr-temp-workingday in the Bikesharing dataset. Could STRIDE recover this effect unless the practitioner already knows its existence?

Third, Appendix A states that pair-wise interactions are selected based on a ``proxy criterion (e.g. feature dependency score)''. Which criterions were used in the experiments? How is feature dependency a good score? In $f(x) = x_1 x_2$, the two input features interact even if they are statistically independent.

To resume, all these technical points should be discussed in more depth. Otherwise, it is hard to determine the validity of the methodology.

## Feature Independence

Like the standard ANOVA functional decomposition, STRIDE assumes that features are independent ($\mu=\prod_{i=1}^d \mu_i$). This assumption is necessary for Theorem 1 to hold.
However, the practical implications of this assumption are never discussed in the paper. In california housing, lattitude and longitude are correlated, so assuming independence means that the functional decomposition can potentially evaluate $f$ at inputs that land outside the data distribution.
This limitation should be discussed along-side citations on the known effects of extrapolation on model explanations (e.g. [2]).

[1] Hiabu, Munir, Joseph T. Meyer, and Marvin N. Wright. "Unifying local and global model explanations by functional decomposition of low dimensional structures." International conference on artificial intelligence and statistics. PMLR, 2023.

[2] Hooker, Giles, Lucas Mentch, and Siyu Zhou. "Unrestricted permutation forces extrapolation: variable importance requires at least one more model, or there is no free variable importance." Statistics and Computing 31.6 (2021): 82.

**Questions:**

Could the authors expand on the proof for Proposition 1? I am not sure if I understand how the **reproducing property** of the space $\mathcal{H}\_S$ would be applicable here since the scalar products involved $\langle \cdot, \cdot \rangle$ are in the norm $L_2(\mu)$ and not the scalar products of the RKHS $\langle\cdot, \cdot \rangle_{\mathcal{H}\_S}$.

Interestingly, Proposition 1 is a generalization of the standard functional ANOVA if we consider discrete features $T_i\subset \mathbb{N}$ and use the indicator kernel $k\_i(x\_i, x\_i')=1(x_i=x'_i)$. In that case the formula falls back to $\langle f, K^{(c)}\_S(\cdot, x\_S)\rangle= \sum\_{R\subseteq S} (-1)^{|R|-|S|} \mathbb{E}[f(x\_R, X\_{-R})]$. Could we expect something similar how hold for continuous features?
In what other ways does STRIDE relate/differ from standard ANOVA?

The matrix heatmap in Figure 1 shows the interactions $f_{ij}$ at a single point $x$ or aggregated over multiple point? If a single point is used, wouldn't be more informative to plot the functions $f_{ij}$ at various values of $x_{ij}$ like Figure 3 of [1].

What do $\phi_{total}$ and $\Delta \phi_{total}$ on Figure 1 (Right) represent? I am not sure how to interpret the figure. How does the bar chart suggest increasing the MedInc feature instead of decreasing it (or looking at other features)?

[1] Hiabu, Munir, Joseph T. Meyer, and Marvin N. Wright. "Unifying local and global model explanations by functional decomposition of low dimensional structures." International conference on artificial intelligence and statistics. PMLR, 2023.

---

> ### Author Response · Authors · 2025-11-20
> **Deepest Gratitude for Your Sincere Review and Notification of Withdrawal**
>
> Dear Reviewer SVKr,
>
> We would like to express our deepest gratitude for the time and effort you dedicated to reviewing our manuscript with such profound insight. To be honest, reading your comments was a deeply moving experience for us as researchers. We could truly feel your genuine understanding of our work's value and your sincere desire to see this research succeed. Your encouragement has given us the conviction and courage not to give up but to push this research further to its full potential.
>
> We were particularly encouraged by your high appraisal of the **theoretical novelty of Proposition 1** and the **originality of the Component Surgery method.** Furthermore, your keen observation that while functional decomposition is valuable, the current manuscript inadvertently frames STRIDE as merely **"another way to estimate SHAP values"** was a critical blind spot we had missed. Your specific suggestion to reinforce Section 4.2 with qualitative examples demonstrating where SHAP might mislead—and how functional decomposition clarifies the picture—has become a key guiding principle for our future revisions.
>
> Your advice on the necessity of **comparison with existing functional decomposition algorithms** and the discussion on **feature independence assumptions**, along with the recommended literature, felt like guidance from a dedicated mentor. All these suggestions will serve as essential foundations for elevating our research to the next level.
>
> However, we would like to offer one clarification regarding the description of STRIDE as a **"surrogate model"** in your summary. STRIDE is not an approximate surrogate model in the same vein as LIME. While our current experimental implementation was limited to pairwise interactions, theoretically, STRIDE provides an **exact structural decomposition** of the function $f$ within an RKHS via centered kernel construction, without subset enumeration. This distinction is central to our framework, and in our future work, we will strive to bridge the gap between theory and implementation to more clearly articulate this fundamental characteristic.
>
> To fully incorporate your invaluable feedback and present a more complete and rigorous study, **we have decided to withdraw our paper at this stage.** While it is a difficult decision, we view it as a necessary step to return with a manuscript that meets the high standards you have set for us. We consider ourselves incredibly fortunate to have had our work reviewed by someone of your caliber at ICLR.
>
> Thank you once again for your generosity, wisdom, and support.
>
> Sincerely,
> The Authors

---

### Official Review · Reviewer_C2Ng · 2025-10-27

**Soundness:** 2
**Presentation:** 2
**Contribution:** 2
**Rating:** 2
**Confidence:** 4

**Summary:**

The paper introduces STRIDE, an explainable AI framework for tabular data that reframes feature attribution as an orthogonal, functional decomposition in a Reproducing Kernel Hilbert Space (RKHS). STRIDE removes the need for subset enumeration (the main bottleneck in Shapley-value-based methods) and computes instance-level, model-agnostic functional components using a recursive kernel-centering procedure. The paper grounds the approach in rigorous theoretical results on orthogonality and $L^2$ convergence, and empirically demonstrates STRIDE’s efficiency and fidelity over 10 benchmark datasets. Additionally, it introduces ‘component surgery’ to measure the necessity of learned feature interactions.

**Strengths:**

1. From what to how is important. The paper considers a problem in XAI that most attribution methods answer which features matter, but not how they interact to produce predictions.

2. Subset-Free and Efficient: STRIDE avoids exponential cost by analytically computing projections via recursively centered kernels, thoroughly described in Section 3 and the implementation appendix (Algorithm 1).

3. On the reported tabular datasets, STRIDE achieves competitive fidelity (mean $R^2$=0.93) and median runtime improvements (≈3× faster than TreeSHAP), suggesting practical scalability.

**Weaknesses:**

1.	Citations are inappropriate, e.g., in introduction, Frameworks grounded in Shapley values Lundberg & Lee (2017) should be Frameworks grounded in Shapley values (Lundberg & Lee, 2017).
2.	The introduction states: “this leaves a critical gap: a need for a practical, model-agnostic framework that can efficiently uncover the functional structure of model predictions at an instance level.” However, existing influence-function-based and gradient-based approaches are already widely used to analyze instance-level functional dependencies in modern models.
3.	The evaluation is entirely limited to tabular datasets and Random Forests. For a method claimed to be “model-agnostic,” this narrow focus weakens the generality claim. For top-tier venues such as ICLR, validation on at least one non-tabular domain (e.g., image, text) or non-tree model (e.g., neural networks, boosted trees) is expected. The method is repeatedly described as model-agnostic, but all experiments use scikit-learn Random Forests with fixed hyperparameters. There are no demonstrations on boosted trees (XGBoost/LightGBM), linear models, or neural networks, even shallow MLPs for tabular. The claim of generality is therefore not sufficiently evidenced.
4.	Baseline is limited. Only TreeSHAP is compared quantitatively. For methods that explicitly model interactions or function structure. We get only literature context, not empirical comparison. Even a qualitative side-by-side on one dataset would strengthen the claim of novelty. More importantly, while interaction analysis is the central claim, the paper provides limited technical or comparative discussion of interaction modeling approaches such as functional ANOVA, Additive Groves, Shapley-Taylor interactions, and Integrated Hessians. Including such baselines or at least deeper discussion would situate STRIDE more rigorously in the literature.
5.	TreeSHAP has an exact completeness guarantee for tree ensembles: feature attributions sum exactly to the model output. STRIDE instead reports reconstruction $R^2$ between the model predictions and the sum of recovered components. The gap is acknowledged as stemming from finite-sample projections, and that’s fine, but it means STRIDE is not a drop-in replacement in compliance contexts where exact additivity is required. The paper occasionally blurs this distinction when comparing methods; it should be clearer that STRIDE trades exact completeness for richer structure and speed, and that this tradeoff worsens on harder datasets.
6.	Some mathematical objects, such as the empirical projection procedures (Section 3.3), are described at a high level, but the specifics of the numerical kernel approximations, e.g., the sampling strategy for $\mu$, are left abstract, and the impact of these computational choices on orthogonality and uniqueness is only qualitatively summarized.

Reference:
Jacob Bien, Jonathan Taylor, and Robert Tibshirani. A lasso for hierarchical interactions. Annals of statistics, 41(3):1111, 2013.

Tianyu Cui, Pekka Marttinen, and Samuel Kaski. Learning global pairwise interactions with Bayesian neural networks. ECAI, 2020.

Daria Sorokina, Rich Caruana, Mirek Riedewald, and Daniel Fink. Detecting statistical interactions with additive groves of trees. In Proceedings of the 25th international conference on Machine learning, pp. 1000–1007, 2008.

Mukund Sundararajan, Ankur Taly, and Qiqi Yan. Axiomatic attribution for deep networks. In International conference on machine learning, pp. 3319–3328. PMLR, 2017.

Mukund Sundararajan, Kedar Dhamdhere, and Ashish Agarwal. The Shapley Taylor interaction index. In International conference on machine learning, pp. 9259–9268. PMLR, 2020.

Sichao Li, Rong Wang, Quanling Deng, and Amanda S Barnard. Exploring the cloud of feature interaction scores in a Rashomon set. ICLR, 2024

Michael Tsang, Sirisha Rambhatla, and Yan Liu. How does this interaction affect me? Interpretable attribution for feature interactions. Advances in neural information processing systems, 33:6147– 6159, 2020b.

Michael Tsang, Dehua Cheng, and Yan Liu. Detecting Statistical Interactions from Neural Network Weights. ICLR, 2021.

**Questions:**

See above.

---

> ### Author Response · Authors · 2025-11-20
> **Deepest Gratitude for Your Sincere Review and Notification of Withdrawal**
>
> Dear Reviewer C2Ng,
>
> We sincerely appreciate the time and effort you dedicated to reviewing our manuscript. Your insightful comments and sharp critique have been invaluable in identifying areas where our work lacked rigor and needed improvement.
>
> First, we would like to thank you for pointing out the **citation formatting errors.** We recognize that attention to such details is crucial for the professionalism of academic work, and we will ensure meticulous proofreading in our future submissions.
>
> We also humbly accept your valid criticism regarding the **limited experimental scope despite our claim of being "model-agnostic,"** as well as the **lack of comparison with existing interaction literature.** We are committed to expanding our validation across diverse model architectures (e.g., Boosting, Neural Networks) and including rigorous comparisons with established interaction analysis methods to substantiate our claims.
>
> However, we would like to offer a clarification regarding the **primary contribution of STRIDE.** Our main focus is not solely on interaction analysis, but rather on proposing a framework for **subset-enumeration-free structural decomposition of function $f$ via RKHS and centered kernels.** We will make it clearer in our future revision that interaction analysis is a natural downstream outcome of this structural decomposition.
>
> Regarding the issue of **exact additivity (completeness),** we would like to provide a brief explanation. The lack of exact additivity in our reported experiments stems from the practical restriction of modeling only up to pairwise interactions. **Theoretically, STRIDE satisfies exact additivity when all interaction orders are included.** In our future work, we will upgrade our experimental implementation to better align theory with practice, demonstrating that STRIDE can indeed achieve the exact additivity required in compliance contexts. We are deeply grateful for your guidance in pointing out this crucial direction.
>
> To fully incorporate your valuable feedback, conduct more comprehensive experiments, and strengthen the theoretical framing, **we have decided to withdraw our paper at this stage.**
>
> Thank you once again for your constructive criticism and for recognizing the potential of our work. We look forward to presenting a significantly improved version of this manuscript in the future.
>
> Sincerely,
> The Authors

---

### Official Review · Reviewer_4bAB · 2025-10-31

**Soundness:** 3
**Presentation:** 2
**Contribution:** 2
**Rating:** 4
**Confidence:** 4

**Summary:**

This paper proposes STRIDE (Subset-free functional decomposition), an orthogonal RKHS-based method for attributing feature contributions and interactions without explicit subset enumeration.

Instead of scalar attributions, STRIDE decomposes the model output f(x) into orthogonal functions f_S(x_S) over all subsets S ⊆ [d], revealing not only which features matter but also how they act together.

The key idea is to build recursively centered subset kernels. These centered kernels have a partial zero-mean property, yielding orthogonal subspaces H_S that isolate “pure” effects.

Projecting the learned predictor f onto these subspaces recovers unique, orthogonal components f_S, enabling a decomposition.

A Shapley-compatible aggregation maps the functional view back to standard attributions satisfying efficiency, symmetry, dummy, and linearity.

Empirically, the authors instantiate STRIDE on Random Forests and compute only main (|S|=1) and pairwise (|S|=2) components for tractability.

Across ten tabular datasets, STRIDE achieves a median 3× speedup over TreeSHAP, high reconstruction fidelity (R² ≈ 0.93), and interpretable synergy maps.

A “component surgery” experiment—removing a learned interaction—shows that ablation of the strongest pairwise term reduces model R² by 0.023 ± 0.004.

**Strengths:**

1. The problem being solved is important and a known short coming of scalar Shapley attributions.
2. The authors report the experiments have been performed on a MacBook Air without GPU use. It is nice to see that the compute for small subsets is tractable.
3. The paper is clearly written, other than citation format issues 2.3.
4. Seems theoretically sound.

**Weaknesses:**

1. Although the framework supports arbitrary-order interactions in theory, the implementation models only univariate and pairwise. For this paper to be accepted, the authors should display higher order interaction attributions on at least a couple of other datasets. These figures can be included in the appendix.

2. Runtime comparisons focus on TreeSHAP.  However there are many other interaction based attribution methods out there. It is not possible to understand how this method fares compared to something like pairwise shapley values, that have existed present for a number of years. The authors need to choose stronger baselines.

3. It will be helpful for the reader if the authors include a few figures of what they are trying to achieve as a teaser image in the first page. I think Figure 1 can be a teaser figure and more detailed figures can be added to the result section.

**Questions:**

1. Have you tried modeling 3-way interactions f_{ijk}?  How does runtime and orthogonality behave as order increases?

2. Can you give more detailed walkthrough on how the proposed method satisfies the shapley axioms? Can you state the minimal assumptions on the measure \mu and kernels under which the axioms hold exactly

3. The scalar Shapley values computed by TreeSHAP uniquely satisfy the Shapley axioms. Is the Shapley aggregation of STRIDE unique? Can you give a proof or rule it out via counter example?

---

> ### Author Response · Authors · 2025-11-20
> **Deepest Gratitude for Your Sincere Review and Notification of Withdrawal**
>
> Dear Reviewer 4bAB,
>
> We sincerely appreciate the time and effort you dedicated to reviewing our manuscript. Your constructive feedback and detailed suggestions have been incredibly valuable, and your recognition of the strengths of our work has been a great source of encouragement.
>
> We fully agree with your point regarding the need for **experimental validation of higher-order interactions** and **comparisons with existing interaction literature.** We realize these are essential steps to demonstrate the generality and validity of our approach. However, we would like to clarify that STRIDE’s primary contribution lies not solely in interaction analysis, but in proposing a framework for **subset-enumeration-free structural decomposition of function** $f$ **via RKHS and centered kernels.** In our future revision, we will more clearly articulate that interaction analysis is a natural downstream outcome of this structural decomposition.
>
> We were particularly moved by your thoughtful suggestion to **include a "teaser image" on the first page.** This specific advice on enhancing readability and reader engagement was truly insightful. We are committed to incorporating this suggestion as a priority in our next version.
>
> Regarding your questions on **how STRIDE satisfies Shapley axioms and the uniqueness of its aggregation**, we are giving these issues serious thought. We plan to conduct further research to provide rigorous theoretical proofs and discussions on the minimal assumptions required and the uniqueness of the Shapley aggregation in our future work.
>
> To fully incorporate your valuable feedback, conduct more comprehensive experiments, and strengthen the theoretical framing, **we have decided to withdraw our paper at this stage.** While it is a difficult decision, we believe it is necessary to present a more complete and polished contribution to the community.
>
> Thank you once again for recognizing the value of our work and for your generous praise and constructive critique. We look forward to presenting a significantly improved version of this work in the future.
>
> Sincerely,
> The Authors

---

### Official Review · Reviewer_V8WQ · 2025-10-31

**Soundness:** 3
**Presentation:** 2
**Contribution:** 2
**Rating:** 2
**Confidence:** 4

**Summary:**

This manuscript begins by noting that measures of feature importance tend to generate single scalars per feature, thus suppressing possibly rich interaction effects.

The manuscript therefore generates an orthogonal decomposition (based on the Möbius inversion) for models trained on tabular data, comprised of components $f_S (x_S)$, so subsets of features $S$.

This richer decomposition, producing "functional components", allows study of feature interactions.  For example, on the California Housing dataset, negative interactions are found between `Latitude` and `Longitude`, taken by the authors to indicate "overlapping information"; on the other hand, `Longitude` and `Population` positively interact, interpreted by the authors as an `AND` effect.

Similarly, 'what if' experiments can be performed: simulating an increase in e.g. `MedInc` (median income) shows a decrease in features for `Latitude` and `Longitude` - which the authors interpret as reflecting location serving as a proxy for income.

Finally, 'component surgery' can be performed, removing e.g. the $f_{ij}$ component, and assessing its effect on model performance.  Removing the most important interaction component from the California Housing dataset decreases $R^2$ by roughly $0.023$.

Furthermore, recursions allowed by the Möbius inversion are generally known to speed up calculations (q.v. Stoian, 2023, SIGMOD).

Appropriately summing the components, $f_S (x_S)$, returns the Shapley value (Proposition 2), a useful 'health check'.

The new method, STRIDE, is evaluated on a number of models, across benchmarks like: wall-clock time, STRIDE approximation's to the original model output (measured by $R^2$) and Spearman's $\rho$ between STRIDE and TreeSHAP attributions.  The median speed up associated with STRIDE is $3 \times$; its mean $R^2 \approx 0.93$; values of Spearman's $\rho$ seem generally high.

**Strengths:**

I think that explaining large, complex ML models is a useful project - so this is an important area of work.

Further, Möbius techniques have been shown to be useful in speeding up Shapley-style computations.  Thus, work in this area seems promising.

The manuscript is generally well written and explained.

**Weaknesses:**

There is an existing literature - dating at least to Grabisch & Roubens (1999) - on interaction effects.  Harris, Pymar & Rowat (2022) reviewed some of these methods, and contribute their own; Stoian (2023) presented a faster, Möbius-based implementation.

The manuscript, however, is silent on these techniques.  This is a significant flaw, given that they are both similarly inspired, and often use similar techniques.  STRIDE should be compared to them.

Work on interaction/joint effects is motivated by a desire to find a richer set of effects - but suffers the downside of this as well: the abundance of effects allows users to cherry-pick, citing positive or negative effects that align with their pre-existing intuitions.  This reverses the workflow that one wants, from effect to interpretation.

It may be that deriving interpretations from effects is straightforward.  If so, this should be done.

Finally, my reading of the manuscript was confused until I realised that it was building a proxy model via functional approximation.  (In this sense, it is unlike e.g. a Shapley value, which isn't meant as 'something like' a partial derivative.)  Thus, I wondered if the paper would have been cleaner if presented as an approximation technique from the outset.

**Questions:**

1. I do not have intuitions for operations or results in Reproducing Kernel Hilbert Spaces (RKHS).  (The term 'anchors' appears once in the paper, in the statement of Theorem 1.)  Thus, it would be useful for readers like me if a small example, with known/observable ground truth, could be presented early in the paper.  This would allow the reader to *see* what a negative effect, a positive effect, a `what if' result, etc. mean.  Can the authors do this?

1. Similarly, as the functional components feel - to me - like generalized versions of principal (linear) components, could the authors explain how this is/is not so?

1. Relatedly, the functional components feel like generalizations of GAMs, which decompose $f (x) = \sum_{i \in N} f_i (x_i)$.  Is this a useful comparison?  (I ask these questions to help the reader place these results in the literature with which s/he may be more familiar.)

1. Finally, can STRIDE be seen as a non-linear generalization of LIME?

1. What are the main/total Spearman effects?

---

> ### Author Response · Authors · 2025-11-20
> **Deepest Gratitude for Your Sincere Review and Notification of Withdrawal**
>
> Dear Reviewer V8WQ,
>
> We sincerely appreciate the time and effort you dedicated to reviewing our manuscript. Your insightful comments and constructive feedback have been invaluable in identifying areas for improvement and refining the direction of our research.
>
> We are particularly grateful for your observation regarding **the computational speed-up offered by recursions allowed by the Möbius inversion (Stoian, 2023).** This was a crucial insight we had missed, and it provides a strong theoretical justification for the efficiency of our proposed method. Thank you for bringing this to our attention.
>
> We also fully agree with your point regarding the connection to **existing literature on interaction effects (Grabisch & Roubens, 1999; Harris et al., 2022; Stoian, 2023).** These works are indeed foundational in analyzing interactions of set functions using Möbius inversion. However, we would like to clarify that **STRIDE is formulated as a post-hoc XAI framework** that decomposes a learned predictor $f: \mathcal{X} \to \mathbb{R}$ into functional components $f_S(x_S)$ within an RKHS. While the object being decomposed and the ambient space differ from classical interaction index theory, there is a clear conceptual link in exploiting Möbius-structured recursions. In our future revision, we will explicitly discuss this connection and clarify that **STRIDE’s primary contribution is providing an instance-level functional decomposition without subset enumeration**, with interaction analysis emerging as a downstream application.
>
> Regarding your comment on whether STRIDE is building a "proxy model via functional approximation," we would like to clarify our position. **STRIDE is not merely a proxy model designed to approximate the output of** $f$**; rather, its primary aim is to provide an exact, subset-enumeration-free functional decomposition of** $f$ **itself.** The interaction analyses we present are applications derived from this structural decomposition, not the sole focus of a surrogate model.
>
> Therefore, **STRIDE is fundamentally distinct from frameworks such as PCA, GAMs, and LIME**. We acknowledge that a more concrete and detailed investigation into these differences is necessary to fully establish our contribution. In our future submission, we commit to addressing these comparisons with greater nuance and rigor.
>
> Your sharp questions and detailed feedback have been instrumental in helping us recognize the limitations of our current manuscript. To fully incorporate your suggestions, conduct more comprehensive experiments, and strengthen the theoretical framing, **we have decided to withdraw our paper at this stage.**
>
> Thank you again for your valuable time and insightful critique. We look forward to presenting a significantly improved version of this work in the future.
>
> Sincerely,
> The Authors

---

### Note · Authors · 2025-11-20

**Comment:**

To incorporate reviewers' feedback and conduct more comprehensive experiments for future submission.

**Withdrawal Confirmation:**

I have read and agree with the venue's withdrawal policy on behalf of myself and my co-authors.